# Influence of Neonatal Sex on Breast Milk Protein and Antioxidant Content in Spanish Women in the First Month of Lactation

**DOI:** 10.3390/antiox11081472

**Published:** 2022-07-28

**Authors:** David Ramiro-Cortijo, Andrea Gila-Diaz, Gloria Herranz Carrillo, Silvia Cañas, Alicia Gil-Ramírez, Santiago Ruvira, María A. Martin-Cabrejas, Silvia M. Arribas

**Affiliations:** 1Department of Physiology, Faculty of Medicine, Universidad Autónoma de Madrid, C/Arzobispo Morcillo, 2, 28029 Madrid, Spain; david.ramiro@uam.es (D.R.-C.); andrea.gila@uam.es (A.G.-D.); santiago.ruvira@estudiante.uam.es (S.R.); 2Food, Oxidative Stress and Cardiovascular Health (FOSCH) Research Group, Universidad Autónoma de Madrid, Ciudad Universitaria de Cantoblanco, 28049 Madrid, Spain; silvia.cannas@uam.es (S.C.); alicia.gil@uam.es (A.G.-R.); maria.martin@uam.es (M.A.M.-C.); 3Division of Neonatology, Hospital Clínico San Carlos, Instituto de Investigación Sanitaria del Hospital Clínico San Carlos (IdISSC), C/Profesor Martin Lagos s/n, 28040 Madrid, Spain; gherranz@gmail.com; 4Department of Agricultural Chemistry and Food Science, Faculty of Science, Universidad Autónoma de Madrid, Ciudad Universitaria de Cantoblanco, 28049 Madrid, Spain; 5Institute of Food Science Research, CIAL (UAM-CSIC), Universidad Autonoma de Madrid, C/Nicolás Cabrera, 9, 28049 Madrid, Spain; 6PhD Programme in Pharmacology and Physiology, Doctoral School, Universidad Autónoma de Madrid, 28049 Madrid, Spain

**Keywords:** antioxidants, neonatal sex, male disadvantage, breast milk, breastfeeding, macronutrients

## Abstract

Breast milk (BM) is the best food for newborns. Male sex is associated with a higher risk of fetal programming, prematurity, and adverse postnatal outcome, being that BM is an important health determinant. BM composition is dynamic and modified by several factors, including lactation period, prematurity, maternal nutritional status, and others. This study was designed to evaluate the influence of sex on BM composition during the first month of lactation, focused on macronutrients and antioxidants. Forty-eight breastfeeding women and their fifty-five newborns were recruited at the Hospital Clínico San Carlos (Madrid, Spain). Clinical sociodemographic data and anthropometric parameters were collected. BM samples were obtained at days 7, 14, and 28 of lactation to assess fat (Mojonnier method), protein (Bradford method), and biomarkers of oxidative status: total antioxidant capacity (ABTS and FRAP methods), thiol groups, reduced glutathione, superoxide dismutase and catalase activities, lipid peroxidation, and protein oxidation (spectrophotometric methods). Linear mixed models with random effects adjusted by maternal anthropometry, neonatal Z-scores at birth, and gestational age were used to assess the main effects of sex, lactation period, and their interaction. BM from mothers with male neonates exhibited significantly higher protein, ABTS, FRAP, and GSH levels, while catalase showed the opposite trend. No differences between sexes were observed in SOD, total thiols, and oxidative damage biomarkers. Most changes were observed on day 7 of lactation. Adjusted models demonstrated a significant association between male sex and proteins (β = 2.70 ± 1.20; *p*-Value = 0.048). In addition, total antioxidant capacity by ABTS (β = 0.11 ± 0.06) and GSH (β = 1.82 ± 0.94) showed a positive trend near significance (*p*-Value = 0.056; *p*-Value = 0.064, respectively). In conclusion, transitional milk showed sex differences in composition with higher protein and GSH levels in males. This may represent an advantage in the immediate perinatal period, which may help to counteract the worse adaptation of males to adverse intrauterine environments and prematurity.

## 1. Introduction

The benefits of breast milk (BM) on maternal and neonatal health have been amply demonstrated. Breastfeeding reduces mortality and morbidity rates in infants [1,2,3,4], being that the relative risk of infant death is 15 times higher in non-breastfed compared to BM-fed infants [5]. Besides, early cessation of breastfeeding has a detrimental impact on infant health in the short- [6,7] and long-term [5,8], related to the BM benefits reducing the risk of chronic non-communicable diseases [9,10]. The capacity of BM to prevent diseases is related to the wide range of bioactive components that it contains—hormones, antioxidants, growth factors and immunoglobulins, and others—which have developmental and immune functions [11,12]. BM composition is influenced by multiple factors, including the lactation period [13,14], maternal nutritional status [13], and neonatal variables such as gestational age [15] and growth [16]. 

Sex exerts an important influence from early life; males in general are at a disadvantage, particularly in adverse situations. Firstly, the placenta exhibits a sexual dimorphism; being female placenta more adaptable [17,18]. The better ability of female placenta to adjust to changes may also explain the worse outcomes of males in situations of insufficient blood, oxygen or nutrient supply evidenced in humans [19,20] and confirmed in experimental animals [21]. A meta-analysis suggests that the occurrence of major pregnancy complications is also higher in women carrying male fetuses [22]. Finally, males also exhibit a higher incidence of prematurity and associated postnatal morbidities [23]. 

Even though the “male disadvantage”—a term coined by Naeye and co-workers in 1971 referring to the higher rates of perinatal mortality for male infants [24]—is now widely recognized, its biological grounds are still not completely understood. One of the proposed hypotheses is a lower degree of maturation in males at birth, particularly related to the antioxidant systems, which is relevant in infants with premature birth [23]. In this context, BM may be relevant due to critical bioactive factors compensating for this deficiency. The importance of taking sex into account for nutrition in infants born under adverse conditions, such as prematurity, has been recently put forward [25]. In this context, several questions arise: is BM different according to sex? Should sex be considered when selecting donated milk? We have evidenced that sex influences maternal milieu during gestation and that a woman carrying a male fetus exhibits a higher pro-inflammatory cytokine profile [26]. Therefore, it is also possible that sex may influence BM composition. However, this is an aspect not sufficiently explored in humans with limited and conflicting results [25,27]. The aim of this study was to increase knowledge of this field, evaluating the influence of sex on BM composition during the first month of lactation and focusing on macronutrients and antioxidants, which are relevant factors in the transition from the fetal to neonatal periods, particularly under conditions of low birth weight or insufficiently developed physiological systems.

## 2. Materials and Methods

### 2.1. Population of Study

This study included mothers who gave birth at the Obstetrics and Gynecology service and those with neonates enrolled during the first 24 h of life at the Neonatal Intensive Care Unit (NICU) of Hospital Clínico San Carlos (HCSC, Madrid, Spain) between October 2019 to March 2020 (the time of stay in the NICU was not collected). All the women participating in the study signed an informed consent. The study was approved by the HCSC Ethical Committee (19/393-E) and followed the Helsinki declarations involving human subjects. Appendix A reports the Strength Reporting of Observational Studies in Epidemiology (STROBE) Statement checklist for cross-sectional studies. 

The inclusion criteria were women with pregnancy without any fetal malformations, chromosomal or metabolic abnormalities, which may affect enzymatic activity, and the maintenance of lactation during the first month postpartum. With these criteria, forty-eight mothers—anonymous and voluntary—were included in the study, 7 of whom had twin pregnancies, and from which 2 were sex discordant twins. Each participant provided a BM sample at three time-points if possible (see below for details). From each woman, the following data were recorded from a sociodemographic questionnaire and medical records: maternal age (years), maternal country of birth (origin), educational level, gravidity, number of previous abortions, assisted reproduction techniques (ART; no/yes), twin gestation (yes/no), C-section (yes/no), and gestational age (weeks). Maternal anthropometry was assessed in the first week postpartum: weight, in kg and height in cm (Seca 217, TAQ Sistemas Médicos, Madrid, Spain) and percentage of body fat and muscle by a bioimpedanciometer (Omron Healthcare HBF-514C Full Body Sensor W Scale, Madrid, Spain). The following neonatal data were recorded from medical records: neonatal sex, Apgar score at 5 min, birth weight (g), length (cm) and head circumference (cm), and calculated Z-score. 

### 2.2. Collection and Processing of Breast Milk Samples

A 5 mL portion of breast milk was obtained by each mother using hand self-expression with an electric breast pump (Symphony^®^ Medela, Barcelona, Spain). The BM samples were collected between 10 a.m. to 12 p.m., always after infant feeding, from both breasts, and then the breast milk was pooled. Collection was performed at days 7 ± 2, 14 ± 2, and 28 ± 2 of the lactation period. We did not collect colostrum samples, for ethical reasons, due to the small volume that can be obtained, reserving it for the neonate. During collection, no fortification or supplementation was added, and the sample was processed in the laboratory within the next 3 h after extraction. The pooled sample was divided into two parts; 1 mL were used for fat determination. The remaining 4 mL were centrifuged 3 times at 2000× *g* for 10 min (4 °C) to remove fat content and to recover the aqueous phase, in order to minimize interferences on the colorimetric assays, as previously described [15]. For the extraction of the aqueous phase, glass serological pipettes were used at each centrifugation step. Thereafter, both fat and aqueous phases were aliquoted and stored at −80 °C until use. All BM samples were analyzed within a month to minimize the loss of antioxidants.

### 2.3. Fat and Protein Content in Breast Milk

**Fat**. BM fat was analyzed in non-centrifuged milk samples by the Mojonnier method [28] with slight modifications. Briefly, 0.5 mL of raw BM was mixed with 0.5 mL of ammonium hydroxide to reduce the acidity and to dissolve protein. A 50 µL volume of ethanolic phenolphthalein solution (0.5% *w*/*v*) was added and the mix shaken. Thereafter, 2.5 mL of ethanol, 2.5 mL of ethyl ether, and 2.5 mL of petroleum ether were added and vigorously shaken for 30 s, followed by centrifugation at 4000× *g* for 3 min at room temperature. The upper phase containing fat was stored. This process was repeated in triplicate, adding to the aqueous phase 1 mL of ethanol, 1.5 mL of ethyl ether, and 1.5 mL of petroleum ether. Thereafter, the fat-containing samples were placed—overnight, uncapped—in a gravity convection oven at 50 °C to ensure complete evaporation of ether solvent. Finally, total fat content was measured by gravimetry and expressed as % (*w*/*v*) of BM. To correct fat measure, a blank reaction was performed by substituting BM volume for 0.5 mL of H_2_O-Q. 

**Protein**. Protein quantification was carried out in the aqueous phase of BM following the Bradford method [29]. Briefly, 10 µL of BM (1:50 *v/v* in H_2_O-Q) were mixed with 200 µL of Coomassie-blue dye (1:4 *v/v* in H_2_O-Q; Bio-Rad Laboratories, CA, USA) in a microplate. Bovine serum albumin (Sigma-Aldrich, St. Louis, MO, USA) was used for the standard curve (range 0.1–0.5 mg/mL). To perform the blank curve, Coomassie-blue dye was substituted by H_2_O-Q. After shaking the mix for 1 min, the absorbance was measured at 595 nm in a microplate reader (Synergy HT Multimode; BioTek instruments, Winooski, VT, USA). Total protein levels in BM were expressed as mg/mL.

### 2.4. Breast Milk Antioxidants

All the colorimetric assays were performed in duplicate. In addition, the methods were checked to avoid turbidity as previously published [15]. 

**ABTS radical scavenging capacity (ABTS)**. Total antioxidant capacity of the samples was assessed by the 2,2’-azino-bis-(3-ethylbenzothiazoline-6-sulfonic acid) radical cations (ABTS^•+^) method, as previously reported [30]. ABTS^•+^ was obtained by reacting 7 mmol/L ABTS solution with 2.45 mmol/L potassium persulfate and stirring it in the dark at room temperature for 16 h before use. The ABTS^•+^ solution was diluted in 5 mmol/L PBS (pH = 7.4; 1:75; *v*/*v*), to set an absorbance of 0.7 ± 0.02 at 734 nm. Then, 30 µL of BM, diluted in methanol (1:40 *v*/*v*), were mixed with 270 µL of ABTS^•+^ solution. The reaction was incubated at 37 °C for 5 min, and the absorbance was measured at 734 nm in a microplate reader (Cytation 5; BioTek; Winooski, VT, USA). Calibration curves were constructed using a standard solution of Trolox (range 0.0–0.06 mg/mL). The blank curve was obtained by substituting the BM sample with phosphate buffer saline (PBS). ABTS antioxidant capacity was calculated as mg Trolox equivalent (TE)/mL.

**Ferric Reducing Antioxidant Power (FRAP)**. FRAP was measured following the method described for plasma [31], adapted to BM samples. In a microplate, 10 µL of BM serum was mixed with 300 µL of FRAP reagent [0.3 M acetate buffer; pH 3.6; 10 mM tripyridyl s-triazine (TPTZ) in 40 mM HCl and 20 Mm FeCl_3_-6H_2_O]. The calibration curve was performed with a standard Trolox solution in the range of 25 to 800 µM. After 10 min of incubation at 37 °C, the absorbance was read at 593 nm in a plate reader. FRAP antioxidant capacity values were expressed as µM Trolox equivalent (µM TE).

**Thiol groups**. Thiol levels in the aqueous phase samples were quantified using Ellman’s reagent 5,5-dithiobis-2-nitrobenzoic acid (DTNB) [32], adapted to a microplate reader [33]. In the microplate, 10 μL of the aqueous phase of BM diluted in PBS (1:2 *v/v* pH = 7.4) was mixed with 200 μL of DTNB (0.5 mM in PBS). The standard curve was performed with L-GSH (range: 0.1–0.5 mM), substituting DTNB by PBS for the blank curve. The plaque was incubated for 30 min at dark room temperature. Then, the absorbance was measured at 412 nm in a microplate reader (Synergy HT Multimode; BioTek; Winooski, VT, USA), and thiol levels were expressed as mM GSH/mL.

**Reduced glutathione (GSH)**. GSH levels in BM were assessed by a fluorometric method based on the o-phthalaldehyde (OPT) reaction [32] adapted to a microplate reader [33]. A 10 μL volume of the aqueous phase of BM was mixed with 12.5 μL of HPO_3_ (25% *w*/*v* in H_2_O-Q) and 37 μL of PBS (0.1 M), followed by incubation for 10 min on ice and centrifugation at 2100× *g* for 20 min at 4 C. In a microplate, a 2 μL sample volume was mixed with 188 μL of PBS (0.1 M) and 10 μL of OPT (10% *p*/*v* in methanol). The microplate was shaken for 1 min and incubated for 15 min in the darkness at room temperature. Fluorescence was measured at 360 ± 40 nm excitation and 460 ± 40 nm emission wavelengths in a microplate reader (Synergy HT Multimode; BioTek; Winooski, VT, USA). The standard curve was performed with L-GSH (range: 0–10 mg/mL) using PBS as the blank. GSH was expressed as mg GSH/mL. 

**Catalase activity**. Catalase activity was assessed as previously described [15,33]. The method is based on the oxidation of the Amplex Red (Amplex ultra red reagent; Invitrogen, ThermoFisher, MA, USA) by hydrogen peroxide in the presence of horseradish peroxidase, and the reduction of fluorescence by the catalase present in the sample. A 2 μL volume of BM samples were diluted in H_2_O-Q (1:50; *v*/*v*). The diluted samples were mixed with 25 μL of H_2_O_2_ (40 μM) and incubated at 37 °C for 15 min. Then, 50 μL of mix reagent were added, which contained: 25 μL of Amplex Red 10 mM, 10 μL of type I peroxidase 100 U/mL and 2.65 μL Trizma^®^-base 1X pH = 7.5 (Sigma-Aldrich, St. Louis, MO, USA). The mix was incubated at 37 °C for 30 min. The standard curve was obtained with bovine liver catalase (U/mg, Sigma-Aldrich, St. Louis, MO, USA; range: 0.026–5 U/mL). Fluorescence was measured at 530 ± 25 nm excitation and 590 ± 35 nm emission wavelengths in a microplate reader (Synergy HT Multimode; BioTek; Winooski, VT, USA). Catalase activity was expressed as U catalase/mg protein. 

**Superoxide dismutase (SOD).** SOD activity was assessed by a kit (SOD Activity Assay kit KB-03-011, Bioquochem, Gijon, Spain) according to the manufacturer′s instructions. Briefly, 20 µL of BM aqueous phase was diluted in H_2_O-Q (1:1 *v*/*v*), 200 µL of the working solution and 20 µL of enzyme solution were added, and the mix was incubated at 37 °C for 20 min. The absorbance was measured at 450 nm in a microplate reader (Synergy HT Multimode; BioTek; Winooski, VT, USA). SOD activity was expressed in % inhibition. 

### 2.5. Oxidative Damage of Breast Milk Lipids and Proteins

**Lipid peroxidation**. Lipid peroxidation was assessed through evaluation of malondialdehyde (MDA) and 4-Hydroxy-Trans-2-Nonenal (HNE) levels, which are stable products. MDA + HNE levels were determined by a kit (Lipid Peroxidation Assay kit KB-03-002, Bioquochem, Gijon, Spain) according to the manufacturer′s instructions. Briefly, 100 µL of BM were mixed with 325 µL of chromophore reagent kit and incubated at 40 ºC for 20 min. 200 µL of the reaction or standard curve samples were transferred to a microplate reader, and the absorbance was measured at 586 nm (Synergy HT Multimode; BioTek; Winooski, VT, USA). MDA + HNE levels were expressed as µM.

**Protein oxidation**. Protein oxidation was assessed through evaluation of the levels of carbonylated proteins, with a dinitrophenylhydrazine (DNPH)-based method [32], adapted to a microplate reader (Synergy HT Multimode; BioTek; Winooski, VT, USA), measuring absorbance at 370 nm as previously described [15,33]. The levels of carbonyl groups were determined using extinction coefficient of 2,4-dinitrophenylhydrazine (ε = 22,000 M/cm) and were expressed as nmol/mg protein.

### 2.6. Statistical Analysis

Statistical analysis was performed with R software (version 3.6.0, 2018, R Core Team, Vienna; Austria) within R Studio interface using *ggpubr*, *devtools*, *rstatix*, *nlme*, *car*, *ranova* and *ggplot2* packages. 

Data were expressed as median and interquartile range [Q1; Q3] for quantitative variables and the relative frequency (%) was used to summarize qualitative variables. To detect differences in proportions, a Chi-squared test was used. Sex differences in quantitative variables were assessed by Mann-Whitney U test adjusting nominal *p*-Value by multiple comparisons using Holm-Bonferroni method. 

Linear mixed models using subject along the day of lactation as random effect were used to study the influence of male sex on BM macronutrients, antioxidants, and oxidative damage biomarkers. The models included main effects for neonatal sex (using female as reference), lactation period (using day 7 as reference) and were adjusted for significant variables in the univariate analysis and variables known to affect the BM biomarkers (maternal body composition and neonatal standardized anthropometric parameters). The interaction between male sex and day was recorded. Coefficients (β) ± standard error and associated *p*-Value were extracted from neonatal sex and lactation period independently, and from the interaction between neonatal sex and lactation period [15]. The Akaike information criterion (AIC) and Bayesian information criterion (BIC) of the models were reported. Significance probability was established at *p*-Value (*p*) < 0.05.

## 3. Results

Regarding sociodemographic data, 60.0% of the women were from Spain, 20.0% from South America, 3.6% from Asia, 3.6% from Central Europe and 1.8% from North America. With respect to the level of studies, 18.2% had middle school education, 30.9% high school degree, 32.7% bachelor’s degree, and 7.3% had completed postgraduate studies. The median of gravidity was 2 [1; 2]. Maternal age was 34.0 [31.5; 37.0] years old, and gestational age was 37.3 [33.8; 39.0] weeks of gestation. Considering neonatal sex, 43.6% of the newborns were male. 

With respect of the rate of prematurity, we found that it was higher in males, being 37.5% term and 62.5% preterm delivery, although it did not reach statistical significance (χ^2^ = 3.84; *p*-Value = 0.050). However, male neonates had significantly lower gestational age than females (Table 1). We did not detect statistical differences in maternal age, obstetric or anthropometric characteristics between women with a male or female infant. Likewise, no statistical differences at birth were found regarding neonatal characteristics, including rate of fetal growth restriction, anthropometric characteristics, and Apgar score. 

### 3.1. Differences in Breast Milk Levels According to Newborn Sex

At day 7 of lactation, there was a trend towards higher levels of fat in male BM, which did not reach statistical significance (*p*-Value = 0.064). No differences between sexes were found in fat content at days 14 and 28 of lactation (Figure 1A). 

At day 7, proteins were significantly higher in male BM compared to female, without statistical differences between sex at 14 and 28 days of lactation (Figure 1B).

The total antioxidant capacity assessed by ABTS was significantly higher in the BM of male infants compared to females at day 7, with no significant differences detected at days 14 and 28 postpartum (Figure 2A). The total antioxidant capacity assessed by FRAP was significantly higher in male BM at day 28, with no significant differences detected at days 7 and 14 (Figure 2B). 

Regarding low molecular weight antioxidants, GSH was significantly higher in male BM at day 7 and day 14, without significant difference at day 28 (Figure 3A). Total thiol groups showed no significant differences between BM from mothers with a male and female newborn in any of the points studied (Figure 3B). 

With respect to enzymatic antioxidants, catalase activity was significantly lower in male BM at day 7, with no significant differences detected at days 14 and 28 postpartum (Figure 3C). SOD activity did not show significant differences between sexes (Figure 3D).

No differences were detected between males and females in the levels of MDH+HNE or carbonylated proteins, which are biomarkers of oxidative damage to lipids and proteins, respectively, at any of the lactation points studied (Figure 4).

### 3.2. Association between Newborn Sex and Lactation Period in Breast Milk Levels

Considering the main effects of our models (adjusted by maternal weight, body fat in the first week postpartum, gestational age, and neonatal Z-scores), BM protein content was significant and positively associated with male sex.

Total antioxidant capacity analyzed by ABTS method showed a trend towards a positive association with male sex, although it did not reach statistical significance (*p*-Value = 0.056), while no significant association was found between sex and total antioxidant capacity assessed by FRAP. GSH also showed a positive association with male sex, near statistical significance (*p*-Value = 0.064). Regarding catalase activity, a trend towards a negative association with male sex was found, without reaching statistical significance (*p*-Value = 0.078; Table 2). In addition, when gestational age was removed from the models, the effect of neonatal sex remained significant for total antioxidant capacity assessed by ABTS (Appendix A).

None of the interactions between neonatal sex and days of lactation were significant. When the interaction term was excluded in the models (Appendix A), the main effect of male sex on BM proteins was less prominent (β = 1.34 ± 0.72; *p*-Value = 0.074). However, the effect of male sex in BM GSH level remained significant (β = 1.20 ± 0.47; *p*-Value = 0.016). These results indicated that sex and lactation period were independent factors modulating the levels of proteins and antioxidants in BM.

## 4. Discussion

Sex is a relevant factor in perinatal health, being that males are more vulnerable, particularly in situations of low birth weight. Given the important benefits of BM for vulnerable neonates, this study aims to evaluate if neonate sex has an impact on BM composition. We focused our study on macronutrients (fat and proteins) and antioxidants, which are particularly relevant for infants born prematurely, with low birth weight and insufficient degree of maturation of their antioxidant systems. We centered our study in the first month of lactation, since it is the most vulnerable period for survival. Our main findings are that sex is a factor influencing BM protein and antioxidant capacity, with higher levels in milk from mothers with a male infant. The higher macronutrient and antioxidant content in male BM may represent a benefit counteracting “male disadvantage” in the immediate neonatal period. Therefore, the influence of sex on maternal milk should be considered for infant nutrition during this vulnerable period and infants born under adverse conditions.

Male sex is associated with higher rates of prematurity and worse adaptation to adverse intrauterine environment. Although the precise mechanism remains unclear, this has been related to the influence of sex hormones, sensitivity to inflammation and differences in organ maturation [23,34]. Our data also evidence this tendency towards higher rates of prematurity in males. In this context, nutrition during immediate postnatal period may be a critical factor for survival, and it has been proposed that sex differences in BM composition should be considered to optimize nutrition [25]. Information about the numerous factors which affect BM composition has been gradually gathering. However, the effect of sex has been insufficiently explored and focused mainly on macronutrients. Studies in animal models evidence that BM from males has higher energy and protein content [27]. However, human studies have yielded conflicting results. In American well-nourished women, the trend is similar to that found in animals, i.e., BM of male’s mothers exhibited higher caloric content compared to that of females [35]. Higher caloric content in male BM has also found in Korean mothers [36]. Our study, evidencing a higher protein and a tendency towards a higher fat content in BM from women with a male infant, point in the same direction. Another plausible factor could be that, although mothers were told to obtain breast milk at the same time of the day, and after infant feeding, we could not control for hind and foremilk precisely, which is known to influence fat content [37]. Additionally, we did not collect 24 h BM pool samples, which could have affected the fat levels of the BM. Fujita et al., in a Kenyan population, found that sex differences in BM dependent on the nutritional status of the mothers, i.e., higher fat content in male BM was found in women with higher socioeconomic level, while mothers with poor status -and possibly nutrition- had higher fat level when carrying a girl [38]. The authors explain these results in terms of the Trivers–Willard hypothesis, which predicts the differential parental investment between daughters and sons, depending on maternal condition and offspring reproductive potential [39]. Higher energy intake of mothers carrying a male infant has been previously shown in EE.UU., evidencing a 10% higher energy consumption in mothers of boys compared to those of girls [40]. We did not evaluate food intake in this population and, therefore this is a possible explanation. Other studies also suggest the influence of maternal body composition or hormonal status [41]. We did not evaluate hormones in BM, but the higher BM content in proteins was evidenced after correction for maternal and neonatal anthropometric parameters. Besides behavioral, nutritional, and anthropometric aspects, a third plausible biological factor may be the influence of sex hormones during pregnancy on breast growth glandular development, which may influence milk composition. Powe et al., found a correlation between change in breast size during pregnancy and energy of the BM, with a tendency towards larger enlargement in breasts from women carrying a male fetus [35]. This aspect deserves further exploration, perhaps in experimental animals, where the characteristics of the glandular system of breasts and BM can be evaluated under experimental conditions mimicking factors known to affect BM composition in humans. 

Sex differences in protein content were only observed in the first week of lactation. We could not evaluate colostrum for ethical reasons. Colostrum is known to contain the highest levels of protein content and, thereafter, protein gradually decreases, along with lactation [42,43,44]. We suggest that it is possible that the higher protein level observed in male BM at day 7 of lactation may also occur for colostrum, an aspect that deserves further confirmation. We propose that the larger protein content in the BM of mothers with male infants in the immediate postnatal period may provide an advantage under situations of deficient growth, such as prematurity or low birth weight, which are more prevalent in males. In preterm neonates, fat-free mass is lower compared to term infants [45] and neurodevelopment is associated with increases in this variable [46]. 

Together with macronutrients, essential for neonatal growth, in the immediate neonatal period, BM content in antioxidants is relevant to provide antioxidant defense systems in the period of adaptation to an oxygen environment. Therefore, we assessed possible sex differences in antioxidants and biomarkers of oxidative damage, an important contributor of pathology during this period in vulnerable neonates [47]. BM contains several antioxidants, including GSH, vitamins, melatonin, and enzymes, among others. They control redox state through both direct scavenging capacity of reactive oxygen species (ROS) and modulating the activity of enzymes, such as SOD and catalase [48]. The BM antioxidant capacity is one of the defense mechanisms by which breastfeeding is protecting infants against diseases. Antioxidants are particularly relevant in the immediate neonatal period since they help to counteract the abrupt increases in oxygen, and subsequent rise in ROS. If there is insufficient development of antioxidants, as can occurs for premature infants [49], the immediate postnatal period represents a high risk to develop oxidative stress related pathologies, such as retinopathy, bronchopulmonary dysplasia, and necrotizing enterocolitis. For these infants, BM may represent an important source of antioxidants [50]. BM antioxidant levels are highest in colostrum decreasing progressively during the lactation period. This has been confirmed in populations of different origin [51,52], including in Spain [15]. This higher antioxidant levels in early life represents an advantage helping vulnerable infants with insufficiently developed antioxidants. The present study evidences a higher content in several antioxidants in BM of the mothers with male infants, including GSH, one of the most relevant BM antioxidants [53]. The same trend was evidenced for total antioxidant capacity assessed by ABTS. On the other hand, total antioxidant capacity by FRAP methodology demonstrated a larger content in male BM at day 28, but not in the first week postpartum. Both methods measure the total antioxidant capacity and free radical scavenging activity, through different technology. ABTS is an inhibition method: A sample is added to a free radical generating system and the inhibition by the sample is related to the antioxidant capacity of the sample [54]. On the other hand, FRAP method evaluates the iron reducing capacity of a sample [31]. Therefore, they evaluate antioxidants with different chemical mechanisms of action and therefore, their values may reflect different molecules present in the sample. For example, antioxidant capacity assessed by ABTS correlates with GSH [55]. Our results in human milk samples confirm that GSH and ABTS follow the same trend along lactation period and suggest they may be associated. On the other hand, FRAP has been shown to correlate with plasma levels of ascorbic acid and α-tocopherol [31]. The observed larger FRAP found at day 28 in male BM may suggest that at this period of lactation for other antioxidants, such us vitamins, may be more prevalent. Taken together, our data evidence the dynamic changes in BM antioxidants along lactation period. 

Enzymatic antioxidants are also important to counteract oxidative processes. SOD is the first enzyme involved in the detoxification of superoxide anions, while catalase is involved in the second step, removing hydrogen peroxide. Both SOD and catalase have been found in human BM [50]. How the enzymatic antioxidant reaches the milk is not completely understood. It is possible that they are present in the mammary gland and exported to the milk through the membrane of the milk fat globule [56]. In the neonatal period, SOD has been shown to be lower in preterm male infants compared to female counterparts [23], a trend also found in adults and aged subjects and confirmed in experimental animals [57,58]. It has been put forward that a high SOD activity in BM would be relevant mechanism protecting neonates from excess superoxide anion and its secondary oxidizing by-products, such as peroxynitrite [23]. We did not find sex differences in SOD activity in BM, suggesting that it may not contribute to the observed sex differences in the total antioxidant capacity of BM. 

Overall, our data indicate a higher antioxidant capacity of male BM. However, catalase activity showed the opposite trend. Other authors have found the activity of catalase was increased with the period of lactation and they postulated the decline in protein content as potential explanation [59]. Since we assessed catalase activity referred to BM protein content, this is a possible explanation for the opposite trend regarding sex differences in this enzyme. However, we want to add another point of view. Catalase activity is susceptible to oxidizing actions of superoxide anion [58]. If superoxide anion production is high, catalase activity may be subsequently reduced. We could not directly measure ROS, since they are short live molecules and only their stable products of oxidation (carbonyl groups or lipid peroxidation products) can be accurately assessed. Despite the higher antioxidant capacity of male BM, we did not detect lower level of oxidative damage biomarkers (proteins and lipids) as expected. Therefore, it is possible that women with a male infant likely generate higher levels of ROS by mammary gland lactocytes [60], which could explain the lower catalase activity and the lack of differences between sexes in the level of oxidative damage. Regarding the consequences of a lower catalase activity in males, we suggest it may have a double-edged sword effect. On one hand, an important antioxidant activity would be reduced. However, it must be considered that catalase eliminates efficiently hydrogen peroxide (H_2_O_2_), which is a mild oxidant, and only in high doses can be detrimental [61]. Besides, H_2_O_2_ is a relevant antimicrobial compound in BM, especially high one week after delivery [62,63], conferring greater protection of newborns from infection and mothers from mastitis [60]. 

The sex differences in antioxidants were analyzed considering possible confounding factors. The small sample size may have influenced the significant association of the analysis. Therefore, it would be important to confirm these data in a larger population. Additionally, it would be necessary to consider that multiple testing could overstate the reported significance level in the univariate analysis. 

The mechanisms driving sex-specific breast milk synthesis are currently unclear and need to be investigated. A possible explanation may be the influence of sex hormones on the mammary gland. Not only female sex hormones, but also androgens play a regulatory role of female reproductive function, including mammary gland development. Androgens, their enzymatic systems, and receptors have been found in the mammary gland [64]. Some data has suggested that androgens may suppress BM production [65], although this data has been contradictory by in vitro studies [66]. Among other actions of androgens and their receptors, they may modify mammary gland epithelial cell proliferation and induce β-casein gene [66]. Therefore, it is possible that they may influence BM components.

## 5. Conclusions

The breast milk of women with male infants have a higher content of proteins and antioxidants, except for catalase activity, independent of maternal and neonatal characteristics and considering gestational age as a key factor. These sex differences in breast milk composition were observed at the beginning of lactation and could decline in mature milk.

The higher content of proteins and antioxidants may benefit male neonates with deficient growth and immature antioxidant systems. The lower content in catalase may also benefit weaker infants with poor immune defenses due to the antimicrobial actions of hydrogen peroxide. Breast milk sex differences should be considered for nutritional intervention protocols and health care, especially in the most vulnerable neonates such as premature infants.

## Figures and Tables

**Figure 1 antioxidants-11-01472-f001:**
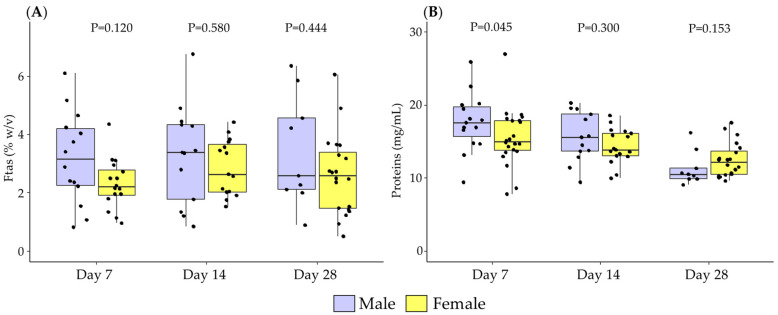
The evolution of breast milk fat (**A**) and proteins; (**B**) during the first month of lactation according to the sex of the newborn. Data show median and interquartile range [Q1; Q3]. Neonatal sex was compared by day using the Mann–Whitney U test and extracted *p*-Value (*p*) adjusted by Holm-Bonferroni multiple comparison methods was reported in the figure.

**Figure 2 antioxidants-11-01472-f002:**
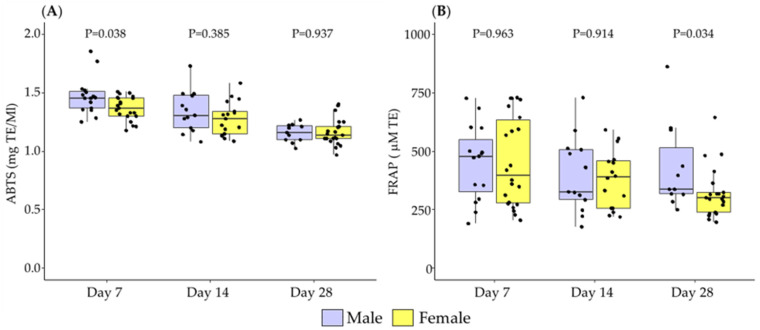
The evolution of breast milk total antioxidant capacity via the ABTS method (**A**), and by FRAP method (**B**) during the first month of lactation according to the sex of the newborn. Data show median and interquartile range [Q1; Q3]. Neonatal sex was compared according to the day of lactation using the Mann-Whitney U test and extracted *p*-Value (*p*) adjusted by Holm-Bonferroni multiple comparison methods was reported in the figure. TE: Trolox equivalent.

**Figure 3 antioxidants-11-01472-f003:**
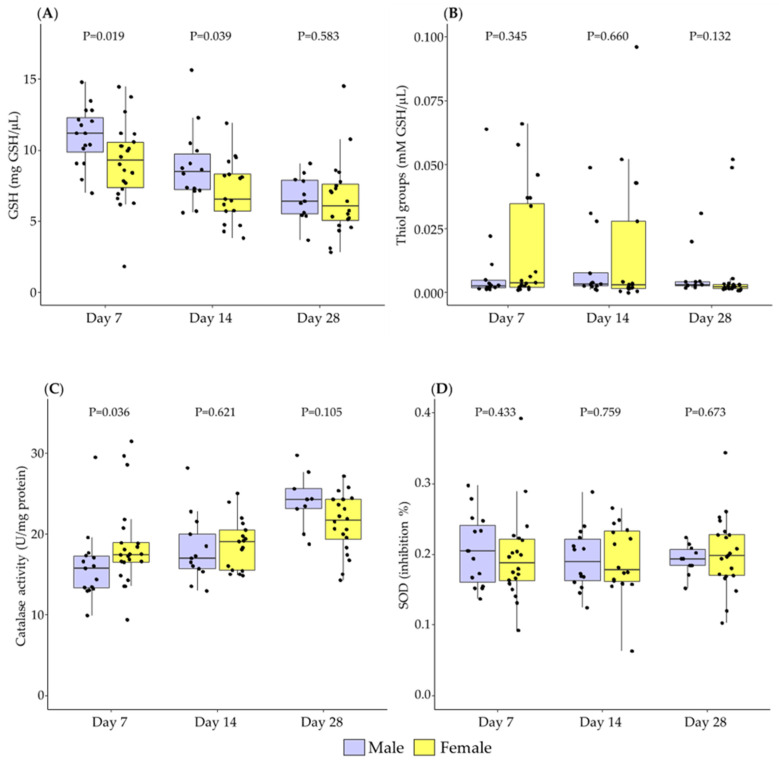
The evolution of breast milk total GSH levels (**A**), thiol groups (**B**), catalase activity (**C**), and SOD activity (**D**) during the first month of lactation according to the sex of the newborn. Data show median and interquartile range [Q1; Q3]. Neonatal sex was compared according to the day of lactation using the Mann-Whitney U test and extracted *p*-Value (*p*) adjusted by Holm-Bonferroni multiple comparison methods was reported in the figure. GSH: reduced glutathione; SOD: Superoxide Dismutase.

**Figure 4 antioxidants-11-01472-f004:**
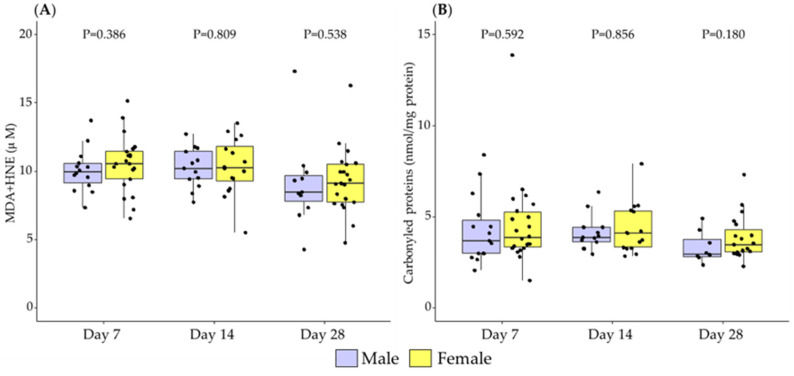
The evolution of breast milk lipid peroxidation (**A**) and protein oxidation (**B**) during the first month of lactation according to the sex of the newborn. Data show median and interquartile range [Q1; Q3]. Neonatal sex was compared according to the day of lactation using the Mann-Whitney U test and extracted *p*-Value (*p*) adjusted by Holm-Bonferroni multiple comparison methods was reported in the figure. MDA: malondialdehyde; HNE: 4-Hydroxy-Trans-2-Nonenal.

**Table 1 antioxidants-11-01472-t001:** Maternal and neonatal characteristics according to neonatal sex; data from Spain between 2019–2020.

	Male (*n* = 24)	Female (*n* = 31)	*p*-Value
Maternal age (years)	34.0 [30.2; 36.2]	35.0 [33.0; 37.0]	0.213
Gestational age (weeks)	34.5 [28.7; 38.3]	38.0 [36.2; 39.6]	0.032
Gravida	2.00 [1.00; 2.00]	2.00 [2.00; 2.00]	0.265
Previous abortion	6 (25.0%)	11 (35.5%)	0.589
Maternal weight (kg)	70.0 [63.1; 81.4]	68.8 [57.1; 76.2]	0.346
Maternal height (cm)	162 [160; 166]	160 [157; 163]	0.178
Body fat (%)	38.2 [36.0; 42.2]	37.8 [32.2; 42.6]	0.651
Muscle (%)	26.1 [24.4; 27.1]	26.3 [25.0; 28.1]	0.492
Twin	4 (16.7%)	3 (9.7%)	0.686
TRA	0 (0.0%)	3 (10.3%)	0.254
C-section	5 (20.8%)	11 (35.5%)	0.375
Fetal growth restriction	2 (9.1%)	4 (12.9%)	1.000
Birth weight (g)	2060 [1280; 3248]	2850 [1735; 3210]	0.594
Birth weight Z-score	−0.71 [−1.36; −0.32]	−0.66 [−1.38; −0.22]	0.999
Birth length (cm)	41.0 [37.4; 49.6]	48.0 [42.5; 48.5]	0.405
Birth length Z-score	−0.99 [−1.28; −0.62]	−0.27 [−1.41; 0.35]	0.396
Birth head circumference (cm)	29.5 [27.0; 34.6]	33.5 [30.1; 34.5]	0.553
Birth head circumference Z-score	−0.39 [−0.80; 0.19]	−0.47 [−1.57; 0.06]	0.686
Apgar at 5 min	10.0 [9.00; 10.0]	10.0 [9.00; 10.0]	0.765

Data show median and interquartile range [Q1; Q3] for quantitative variables and sample size and relative frequency (%) for qualitative variables. The *p*-Value was extracted by Mann-Whitney U test adjusted by Holm-Bonferroni multiple comparison methods in quantitative variables or χ^2^ in qualitative variables.

**Table 2 antioxidants-11-01472-t002:** Linear mixed models with random effects associated with macronutrients and antioxidants of the breast milk.

Main Effects	Proteins	*p*	ABTS	*p*	FRAP	*p*	GSH	*p*	Catalase Activity	*p*
Male	2.70 ± 1.20	0.048	0.11 ± 0.06	0.056	24.82 ± 74.90	0.743	1.82 ± 0.94	0.064	−3.99 ± 2.17	0.078
Day 14	−1.09 ± 1.22	0.383	−0.10 ± 0.05	0.045	−26.87 ± 56.29	0.637	−2.17 ± 0.87	0.019	−0.17 ± 1.50	0.913
Day 28	−3.42 ± 1.40	0.022	−0.20 ± 0.04	<0.001	−62.85 ± 58.03	0.289	−2.53 ± 0.89	0.008	3.19 ± 1.98	0.119
Male: Day 14	−2.39 ± 2.08	0.263	−0.06 ± 0.08	0.410	4.81 ± 89.65	0.958	−3.37 ± 1.41	0.799	3.53 ± 2.56	0.181
Male: Day 28	−0.36 ± 2.48	0.887	−0.12 ± 0.07	0.103	44.91 ± 94.76	0.640	−1.43 ± 1.44	0.332	0.71 ± 3.49	0.841
*AIC/BIC*	*332.2/368.5*		*6.1/43.5*		*762.0/799.5*		*318.8/356.6*		*372.8/409.2*	

Data show coefficients ± standard error and *p*-Value (*p*) associated. In the models, female, and day 7 were considered as a reference. All models were adjusted by maternal weight, body fat in the first week postpartum, gestational age, and neonatal Z-scores of birth weight, length, and head circumference. ABTS: 2, 2’-Azino-Bis-3-Ethylbenzothiazoline-6-Sulfonic Acid; FRAP: Ferric Reducing Antioxidant Power; GSH: reduced glutathione; AIC: Akaike information criterion; BIC: Bayesian information criterion.

## Data Availability

The data presented in this study are available on request from the corresponding author. The availability of the data is restricted to investigators based in academic institutions.

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
