# Peer review of "Influence of Neonatal Sex on Breast Milk Protein and Antioxidant Content in Spanish Women in the First Month of Lactation"

_antioxidants, 2022, doi:10.3390/antiox11081472_

Round 1

Reviewer 1 Report

Overall

The objective of the submitted manuscript is to analyze the potential influence of neonatal sex on proteins and antioxidants of breast milk in the first month of lactation. The paper covers the aim of the journal and the subject investigated is of worldwide interest.

The main concern

I have serious doubts as to whether the differences presented and discussed by the authors result from influence of neonatal sex. As presented in Table 1 there is a statistically important difference in gestational age for male (34.5 [28.7; 38.3]) and female (38.0 [36.2; 39.6]) neonates (p=0.032). It is well established that the concentrations of most components of human milk are gestational age related. In fact, the median gestational age for male neonates classify this group as "late preterm" and can not be compared with "term" female neonates.

Moreover, the Authors missed very important fact that the concentration of many components, in particular fats change over the day and additionally is different at the beginning and at the end of breastfeeding. In Material and Methods, Results and Discussion sections there is no data concerning the type of milk, namely hind and foremilk. The indication of the type of milk is extremely important especially in the determination of fat concentration. Additionally, only 5 ml of milk samples were collected which indicates that the breasts were not completely emptied, which has a significant impact on the obtained results.

The collection of milk samples were done in the wrong way: “A 5 mL of the breast milk (if possible) was obtained from both breasts by an electric  breast pump (Symphony® Medela, Barcelona, Spain), between 10 a.m. to 12 p.m.”

Failure to take into account these differences may affect the final conclusions.

Minor

The details concerning the methods used for macronutrients and antioxidants determinations should be provided.

Additionally, extensive editing of English language and style are required.

Line 38 and further throughout the manuscript

Transitional BM from males

Line 78 Evidence from animal models support this hypothesis.”

The details and appropriate references should be provided

Line 190-191 200 µL of the reaction or standard curve samples were transferred to a microplate reader”.

The details should be provided

Figure 1. The concentration unit for fat is incorrect.

Line 365 Sex is a relevant factor influencing gestational, fetal, and neonatal health.

The statement should be soften up.

Line 370 We centered our study in first month of lactation, since it is a critical period for survival.”

The statement should be soften up.

Author Response

Overall. The objective of the submitted manuscript is to analyze the potential influence of neonatal sex on proteins and antioxidants of breast milk in the first month of lactation. The paper covers the aim of the journal and the subject investigated is of worldwide interest.

Response: Thank you for taking your time to review the article. The authors have responded your comments and modified the main text according to your suggestions.

The main concern. I have serious doubts as to whether the differences presented and discussed by the authors result from influence of neonatal sex. As presented in Table 1 there is a statistically important difference in gestational age for male (34.5 [28.7; 38.3]) and female (38.0 [36.2; 39.6]) neonates (p=0.032). It is well established that the concentrations of most components of human milk are gestational age related. In fact, the median gestational age for male neonates classifies this group as "late preterm" and cannot be compared with "term" female neonates.

Response: This is an interesting comment, and we strongly agree that most of the breast milk macronutrients, such as protein, are influenced by gestational age. However, there is less evidence regarding bioactive factors, such as antioxidants and our previous data demonstrate that some, but not all, BM antioxidants are modified by gestational age (doi:10.3390/nu12092569). In fact, the models were adjusted for confounding variables including gestational age. Considering your comment, we have now reanalyzed the models without adjusting for gestational age, in order to determine if the main effect is changed. Besides, we have modified the models using a linear mixed model with random effect, according to reviewer 4, an expert in statistics. The new analysis demonstrates that the main effect was similar, which demonstrates that the effect of sex is independent of gestational age (Table 2 and Table S2).

Moreover, the Authors missed very important fact that the concentration of many components, in particular fats change over the day and additionally is different at the beginning and at the end of breastfeeding. In Material and Methods, Results and Discussion sections there is no data concerning the type of milk, namely hind and foremilk. The indication of the type of milk is extremely important especially in the determination of fat concentration. Additionally, only 5 ml of milk samples were collected which indicates that the breasts were not completely emptied, which has a significant impact on the obtained results.

Response: We agree with this comment. The macronutrients, particularly the breast milk fat levels, change depending on the time of the day, even within the same intake. Although we collected breast milk samples from both breasts, longitudinally (7, 14 and 28 days) and systematically (always 10 am to 12 pm), we did not collect samples at different times of the day, and we did not control the collection of hind and foremilk. This is a limitation of the study, now explicitly indicated in the discussion (lines 410-413). Regarding to the volume collected (5 mL), it is important to consider that, for ethical reasons, the neonate was first fed, and the remaining milk was used as a sample for the study. We have added this information to the material and methods (section 2.2).

The collection of milk samples were done in the wrong way: “A 5 mL of the breast milk (if possible) was obtained from both breasts by an electric breast pump (Symphony® Medela, Barcelona, Spain), between 10 a.m. to 12 p.m.”

Response: We have modified the collection milk protocol to make it more comprehensible.

Failure to take into account these differences may affect the final conclusions.

Response: Thank you for your comments, we have modified the conclusions taking into account your comments.

Minor

  • The details concerning the methods used for macronutrients and antioxidants determinations should be provided.
  • Additionally, extensive editing of English language and style are required.

Response: The methods have been extended. In addition, the English was extensively reviewed.

Line 38 and further throughout the manuscript: “Transitional BM from males”

Response: The line was edited.

Line 78 “Evidence from animal models support this hypothesis.” The details and appropriate references should be provided.

Response: We have now modified the sentence.

Line 190-191 “200 µL of the reaction or standard curve samples were transferred to a microplate reader”. The details should be provided.

Response: More details of the methods were implemented.

Figure 1. The concentration unit for fat is incorrect.

Response: According to Mojonnier method used, total fat in milk was determined in g/mL. We have now extended the information about this method.

  • Line 365 “Sex is a relevant factor influencing gestational, fetal, and neonatal health.” The statement should be soften up.
  • Line 370 “We centered our study in first month of lactation, since it is a critical period for survival.” The statement should be soften up.

Response: The statements were modified.

Reviewer 2 Report

In this work by Ramiro-Cortijo et al., the authors explore the general milk composition in relation to infant sex. It appears that early milk from mothers that gave birth to males differs in abundance of proteins and antioxidants.

I am an expert in a specific milk component (extracellular vesicles), which I have been studying for several years. Normally I don’t focus on the parameters described in this manuscript. However, I need to know about general milk composition and also proper experimental design to conduct and interpreted my own research in light of others. Therefore, I acknowledge the proper experimental design of this work. I really enjoyed reading this manuscript as it is well written and everything is properly explained (for instance in the methods section). As a milk researcher, I find the data very interesting and it has prompted me to reconsider the source of milk in my own research. Obviously, this work will also be of interest to other researchers, but also clinicians and the general public I believe.

I complement the authors on this interesting work and I support the publication of this manuscript without any reservations.  

Author Response

Response: Thank you very much for taking the time to review the article. In addition, the authors would like to thank you for these kind words about our work.

Reviewer 3 Report

The manuscript "Influence Of Neonatal Sex In Proteins And Antioxidants Of Breast Milk Composition In The First Month Of Lactation" is an interesting scientific work on breast milk composition of male and female infants in the first month of lactation. The work is original and well structured, giving important novelty to scientific literature. The design of the project is appropriate and the results are significant. The number of patients is low but the results look significant. The statistical analysis is well conducted and the language is acceptable. It represents a valid work and it gives the opportunity to focus the attention on the differences between male and female infants in the breast milk composition, giving the opportunity to have a focused treatment based on sex since the first month of life. 

Author Response

(The authors gave the same response as above.)

Reviewer 4 Report

Review of Antioxidants 1782709

This is a statistical review only.  I am not competent to discuss the substantive issues, except to say that the authors are to be commended for investigating this subject.

Please add location (Spain) and years (2019-2020) to the title and the titles of the tables.

The authors show the results of 64 significance tests.  If the tests were independent, the expected number of p-values below 0.05 would be 3.2.  There are 9.  I do not know enough to determine whether these outcomes can be considered independent (actually, the a priori assumption is that the replicate measure within woman are correlated).  If the outcomes are correlated, the expected number would be either higher or lower than 3.2.

The authors do not describe the number of children/mothers who were recruited from the NICU.  This would seem to be an important variable, but it is not used in the analysis or discussed.  What did they do for sex-discordant twins?)

My major issue is that the proper way to think about this data set is as 55 people with replicated samples (even though they are at different times).  Unless demonstrated otherwise, I would expect that a woman who had high values of something in her breast milk at day 7 would also have high values at day 14 and day 21.  It is not clear to me that any of the R procedures cited (lines 200-202) accommodate replicates properly (or that if they do, the authors took this into account).  Furthermore, many of the variables seem skewed.  Should they be transformed before analysis?  This decision should be based on the usual practice with these variables (assuming that someone in the field has given the issue serious thought).  Using a logarithmic transformation, for example, looks plausible for some of the outcome variables and would lead to an interpretation of the results as fractional differences.

Appropriate models would be mixed models with woman as a random effect, including main effects for male sex, day, the appropriate potential confounders (which I hope can include NICU recruitment), then testing for interactions of male sex and day (which, if “significant,” would imply either a diminution or an increase of the child sex effect over time.  If the interactions are not “significant,” this should be noted in the text and the results from models without the interaction terms presented.  This analysis could also be modeled with generalized estimating equations (GEE), with some loss of power.    Both types of models have less power than models assuming (up to) 165 independent measurements (and they should).  If the authors do not know how to do these analyses, they should find a statistician to help them. 

Necessary before acceptance:

1.        Proper analysis of the data;

2.       Acknowledgment in the text that multiple testing means that the nominal p-values reported overstate the “significance level.”

Author Response

This is a statistical review only. I am not competent to discuss the substantive issues, except to say that the authors are to be commended for investigating this subject.

Response: Thank you very much for taking the time to review the statistical part of the article. In addition, the authors would like to thank you for these kind words about our work.

Please add location (Spain) and years (2019-2020) to the title and the titles of the tables.

Response: Location and years were added to table 1. We have modified the title including the origin of women. However, we would prefer to omit the years in the title, since it will lengthen it and this information is provided in section 2.1. 

The authors show the results of 64 significance tests. If the tests were independent, the expected number of p-values below 0.05 would be 3.2. There are 9. I do not know enough to determine whether these outcomes can be considered independent (actually, the a priori assumption is that the replicate measure within woman are correlated). If the outcomes are correlated, the expected number would be either higher or lower than 3.2.

Response: Thank you for the observation, which made us reconsider some statistical aspects of the study and we hope to have improved it. In this pilot, observational and exploratory study, we did not know a priori the true null hypotheses to adjust the alpha significance. However, considering your comment, we have now adjusted the p-value for multiple comparisons in U-test by Holm-Bonferroni method using rstatix package in R. The new statistical analysis has been included in methods and all tables and figures have been updated according to the new p-values.

The authors do not describe the number of children/mothers who were recruited from the NICU.  This would seem to be an important variable, but it is not used in the analysis or discussed.  What did they do for sex-discordant twins?)

Response: We recruited 48 mothers and 55 children, since there were 7 twins. We have now included this number in methods (section 2.1). We had only 2/7 discordant twin cases and were treated as independent.

My major issue is that the proper way to think about this data set is as 55 people with replicated samples (even though they are at different times).  Unless demonstrated otherwise, I would expect that a woman who had high values of something in her breast milk at day 7 would also have high values at day 14 and day 21.  It is not clear to me that any of the R procedures cited (lines 200-202) accommodate replicates properly (or that if they do, the authors took this into account).

Response: This comment is very appropriate. However, we did not consider this approach, since in our previous published data we have demonstrated that some antioxidants are modified during lactation period (doi:10.3390/nu12092569). Therefore, we do not think it is appropriate to consider the different days of lactation as replicated samples. This led us to perform a cross-sectional analysis to have a picture view by day of the effect of neonate sex on bioactive compounds. However, we have reported the interaction effect between sex and lactation day in the models.

Furthermore, many of the variables seem skewed. Should they be transformed before analysis?  This decision should be based on the usual practice with these variables (assuming that someone in the field has given the issue serious thought). Using a logarithmic transformation, for example, looks plausible for some of the outcome variables and would lead to an interpretation of the results as fractional differences.

Response: Also, this comment is very relevant. Generally, biological variables have a high variability as they depend on factors that are difficult to control.  We the logarithmic transformation in our variables did not improve the skewed of these variables. In addition, this transformation would make the interpretation of data difficult in a clinical setting. Therefore, we applied a univariate analysis using non-parametric tests that would give us more robust results.

Appropriate models would be mixed models with woman as a random effect, including main effects for male sex, day, the appropriate potential confounders (which I hope can include NICU recruitment), then testing for interactions of male sex and day (which, if “significant,” would imply either a diminution or an increase of the child sex effect over time.  If the interactions are not “significant,” this should be noted in the text and the results from models without the interaction terms presented.  This analysis could also be modeled with generalized estimating equations (GEE), with some loss of power. Both types of models have less power than models assuming (up to) 165 independent measurements (and they should).  If the authors do not know how to do these analyses, they should find a statistician to help them.

Response: You are right, and we would like to thank you for the suggestion. We have modified the analysis and text applying mixed models with random effects using the nlme R package. We performed the models with interaction terms to test the sex effect over time (Table 2). In this table we have included the interaction effects which were not statistically significant, since we think this information may be relevant for the clinician. However, considering the reviewer´s comments, the models without the interaction effects were also reported in the text and in Table S3.

The models were constructed as follows.

Considering interaction terms

summary(lme(BM biomarker~sex+day+potential confounders+sex:day, random=~day|id, data=bm, na.action=na.omit))

Excluding interaction terms

summary(lme(BM biomarker~sex+day+potential confounders, random=~day|id, data=bm, na.action=na.omit))

Necessary before acceptance:

  1. Proper analysis of the data;
  2. Acknowledgment in the text that multiple testing means that the nominal p-values reported overstate the “significance level.”

Response: Thank you for your comments and deep strategy review, we have modified the analysis and included this consideration in the text (section 2.6.).

Round 2

Reviewer 1 Report

Unfortunately, the arguments and explanations provided by authors are incorrect.

·         Based on the available data on human milk macronutrients, protein concentration is significantly affected by both gestational age at birth and duration of lactation. In view of the above, the compared groups, in order to exclude the potential influence of other factors, should be correctly selected - in this situation - the gestational age for both compared groups cannot differ significantly, as it is the case in this study.

·         In addition, the information in the title is an overinterpretation, without confirmation in the obtained results (change made after performing new analyzes suggested by the statistician) and misleads the reader. „Influence of Neonatal Sex on Breast Milk Protein and Antioxidant content in Spanish Women in the First Month of Lactation”. In Abstract (line 38-39): „In addition, total antioxidant capacity by ABTS (β=0.11±0.06) and GSH (β=1.82±0.94) showed a positive trend near significance (p-Value=0.056; p-Value=0.064, respectively)”.

·         Moreover, the final conclusion in abstract is incorrect (line 40-41) „In conclusion, transition milk showed sex differences in composition with higher protein and GSH levels in males.” GSH level, as stated by authors in line 39, „showed a positive trend near significance” but is not significant.

·         I repeat my point regarding Figure 1. The concentration unit for fat (g/ml) is incorrect. Do the authors mean the number of grams of fat in 1 ml of milk (no details under Figure 1). Nevertheless, according to the original publication cited by the authors [28. Herreid, E.O.; Harmon, C. A Study of Methods of Obtaining Milk Samples for Estimating Milk Fat by the Mojonnier Method. Journal of Dairy Science 1944, 27, 33–38, doi:10.3168/jds.S0022-0302(44)92561-0.], the concentration is given as a percentage [Mean per cent milk fat"]. To the best of my knowledge, the percentage is the amount of grams of a substance in 100 ml (grams) of solution.

·         Additionally, the method for determining the concentration of fat was not well chosen (other methods are currently available, such as standard Miris HMA). The values obtained by the authors are much higher than the available data in this field (see for example Koletzko B. Human Milk Lipids. Ann Nutr Metab. 2016;69 Suppl 2:28-40. doi: 10.1159/000452819).

·         Important research on the impact of gestational length on human milk glutathione peroxidase activity is not cited in the manuscrypt. [Ellis L, Picciano MF, Smith AM, Hamosh M, Mehta NR. The impact of gestational length on human milk selenium concentration and glutathione peroxidase activity. Pediatr Res. 1990 Jan;27(1):32-5. doi: 10.1203/00006450-199001000-00007]

·         In the revised version of the manuscrypt, there are still some unfortunate terms:

transition milk” (line 40) – it should be transitional milk

male BM exhibited" (line 406) - it should be rather BM of male’s mothers

Reviewer 4 Report

Review of Antioxidants 1728709.v2

Thanks to the authors for their swift revision.  However, it still requires more work.

Now we know that there were in fact only 48 independent women providing samples, and that 2 of them had sex-discordant twins.  How the breast milk of a mother with sex-discordant twins is supposed to provide information about the relation of infant sex to anything related to breast milk is not clear to me.  So there are really only 46 women with usable breast milk samples.  PLEASE REDO THE ANALYSIS WITH THE 46 INDEPENDENT WOMEN WITH ONLY ONE CHILD SEX.  Even here, I have no idea what effect twin-ness may have.

There are many differences between infants in the NICU and those who went home shortly after delivery that may affect breast milk.  For example, amount of body contact with the mother, which might have hormonal effects, which might influence the outcomes of interest.  Based on the silence of the authors, I gather that this variable is not available for individual (anonymized) data points (or maybe not at all, considering that they did not add it to Table 1).

As for whether we consider samples on different days “replicates,” I think this is more an issue of terminology, and we can be happy as long as a reasonable analysis is done.

SPECIFIC COMMENTS

LINE       COMMENT

234-236               “The difference between…”  I am not sure what this means.  Does it mean that the p-values shown in the figures for sex differences at different days were adjusted?  Does it mean that the p-values for the Table 2 results were adjusted?  The p-values shown do not look as if they were seriously adjusted for multiple testing.  This is ok, as long as the paper says that the nominal p-values are shown, AND as long as the discussion only talks about results for which the adjusted p-values are low.

243        “Standardized beta coefficients”—In the first place, “beta” is just mathematicalese for “coefficient,” so they can leave “beta” out.  Furthermore, the term “standardized coefficients” usually refers to coefficients divided by their standard errors, or possibly to coefficients of variables that have been divided by their standard deviations (which makes sense for the continuous predictors).  “coefficient +/- SE” is not a standardized coefficient.  Just say that coefficients are presented with their standard errors.

Table 2 (and other tables) AIC and BIC are only useful to compare models.  Their values have no meaning in themselves.  When interactions include more than one term (as they are here), the analyst needs to compare the model with the interaction terms to that without, NOT the individual p-values for the estimated coefficients of the interaction terms.  Based on the 2 tables not including gestational age but differing in whether they include the sex*day interaction terms (S2 and S3), it seems that several outcomes should have the interaction terms included.

Outcome             AIC with interactions       AIC without interactions               difference               include intx?

Protein                 327.6                                   336.6                                                  9.0                        yes

ABTS                     -4.4                                      -2.4                                                     2.0                        no

FRAP                     766.9                                   779.5                                                  12.6                      yes

GSH                       313.5                                   320.7                                                  7.2                        yes

Catalase act        369.3                                   378.6                                                  9.2                        yes

By the way, are the AIC for ABTS correctly recorded?

When presenting the results of models with interactions the results should be shown separately for each category.  In this case, since sex of the infant is the “exposure,” I suggest:

Male-Female (day 7)

Male-Female (day 14)

Male-Female (day 28)

I do not know how to do this in R, but in SAS these would be computed using “estimate” statements in which the point estimates are (respectively)

Main effect for Male

Main effect for Male + interaction effect of Male*day 14

Main effect for Male + interaction effect of Male*day 28

Note that the day effects are not included, since they are the same for both sexes.

The hard part is computing the standard errors (and that’s why it’s helpful if the software does it for you in some convenient way).

The software should be able to report somehow some numbers that can be massaged into a computation of the intraclass correlation coefficients (a measure of how much values within women are correlated) after accounting for day and other factors.  The relevant numbers would be some measure of within-woman variation and the total variation.  Knowing this would be useful to these (and other) investigators in planning further research.

What does “neonatal z-scores” mean?  Are they some value done at or near birth (and if so, what is the upper bound of time?) and carried forward throughout the 28 day time period?  Are they based on values at each age?  Is it really necessary to include all 3? 

Figures:

Thanks for changing the labels on the vertical axes.

I note that these figures don’t look the same as the ones in the previous version (and that the p-values have changed slightly).  Is this related to 48 vs 55?

TO BE DONE BEFORE PUBLICATION:

1.      Explain why NICU is not included in the covariates

2.      Redo models with 46 women

3.      Include sex*day interactions where appropriate and present as described above.